# Psychosocial and professional burden of Medically Assisted Reproduction (MAR): Results from a French survey

**Blandine Courbiere**[1,2]*, **Arnaud Lacan**[3], **Michael Grynberg**[4], **Anne Grelat**[5], **Virginie Rio**[6], **Elisangela Arbo**[7], **Céline Solignac**[7]

**1** Pôle Femmes-Parents-Enfants–Centre Clinico-Biologique d'AMP, AP-HM La Conception, Marseille, France, **2** CNRS, IRD, Aix Marseille Univ, Avignon Université, IMBE, Marseille, France, **3** Kedge Business School, AMSE, CNRS, EHESS, UMR 7316, Marseille, France, **4** Department of Reproductive Medicine & Fertility Preservation, Hôpital Antoine Beclère, Clamart, France, **5** Centre Mistral, Clinique Pasteur, Guilherand-Granges, France, **6** Collectif bAMP, Association de patients de l'AMP et de personnes infertiles, Quincy sous Sénart, France, **7** Medical Department, Gedeon Richter France, Paris, France

* blandine.courbiere@univ-amu.fr

**Data Availability Statement:** All relevant data are within the manuscript. The anonymized MAR data set has been registered in Mendeley Data https://data.mendeley.com/datasets/6rmxbd56cf/1

## Abstract

### Objective

To evaluate the impact of infertility and Medically Assisted Reproduction (MAR) throughout all aspects of life among infertile women and men.

### Materials and methods

An online survey included 1 045 French patients (355 men, 690 women) who were living or had lived the experience of infertility and MAR. The questionnaire included 56 questions on several domains: global feelings, treatment burden, rapport with medical staff, psychosocial impact, sexual life and professional consequences.

### Results

Respondents had experienced an average of 3.6 (95% CI: 3.3–3.9) MAR cycles: 5% (n = 46) were pregnant, 4% (n = 47) were waiting to start MAR, 50% (n = 522) succeeded in having a live birth following MAR, 19% (n = 199) were currently undergoing ART, and 21% (n = 221) dropped out of the MAR process without a live birth. Satisfaction rates regarding the received medical care were above 80%, but 42% of patients pointed out the lack of information about non-medical support. An important impact on sexual life was reported, with 21% of patients admitted having not had intercourse for several weeks or even several months. Concerning the impact on professional life, 63% of active workers currently in an MAR program (n = 185) considered that MAR had strong repercussions on the organization of their working life with 49% of them reporting a negative impact on the quality of their work, and 46% of them reporting the necessity to lie about missing work during their treatment.

**Funding:** Financial support of the IPSOS Survey and the statistical analysis was provided by Gedeon Richter France. The funder provided support in the form of salaries for C. Solignac and E. Arbo but did not have any additional role in the study design, data collection and analysis, decision to publish, or preparation of the manuscript. The funder also assumed all the expenses for publication's fees. The specific roles of these authors are articulated in the 'author contributions' section.

**Competing interests:** AL and VR report no conflict of interest. BC and MG reports consulting fees and speaker's fees from Gedeon Richter France. AG reports consulting fees from Gedeon Richter France. Gedeon Richter provided salaries for CS and EA. The commercial affiliation with Gedeon Richter France doesn't alter adherence of co-authors to PLOS ONE policies on sharing data and materials. No authors have any competing interests that could be perceived to bias this work. There are no patents, products in development or marketed products associated with this research to declare.

## Conclusion

Despite a high overall level of satisfaction regarding medical care, the burden of infertility and MAR on quality of life is strong, especially on sexuality and professional organization. Clinical staff should be encouraged to develop non-medical support for all patients at any stage of infertility treatment. Enterprises should be warned about the professional impact of infertility and MAR to help their employees reconcile personal and professional life.

## Introduction

Infertility is defined as the failure to achieve pregnancy after at least 12 months of regular unprotected sexual intercourse [1]. Infertility affects approximately 6% of US couples [2] and 10% to 20% of couples in Europe [3,4]. Since pioneering work was published in the early 1980s [5], a number of publications have shown that infertility is associated with many psychological and social consequences [6,7]. For infertile couples, a wide array of psychological issues has been described and measured [8,9], including but not limited to depression, anxiety, sexual dysfunction, and social isolation. Although not always strictly consistent, the vast majority of existing reports have demonstrated that both infertility and Medically Assisted Reproduction (MAR) procedures generate a substantial burden on infertile couples.

The last ESHRE and International Committee for Monitoring Assisted Reproductive Technology (ICMART) reports showed a continuing increase in the use of ART worldwide [10,11]. According to the ICMART data that reported about two-thirds of the worldwide ART activity, 1 643 912 ART cycles led to the birth of more than 394 662 babies in 2011 (data excluding Republic of China). ART activity in China has been estimated at 2 million cycles with 500 000 babies per year. In ART centers, ART outcome criteria such as the live birth rate, multiple pregnancy rate, and ovarian hyperstimulation syndrome rate are closely monitored. Despite some tools as SCREENIVF have been validated for screening patient at risk of emotional distress after ART [12], the psychosocial consequences of ART still seem to be a lower priority as a center quality indicator than effectiveness and safety. In 2015, in ESHRE guideline, Gameiro et al. reported international recommendations for providing routine psychosocial care in infertility and medically assisted reproduction [13]. Psychosocial care could reduce stress about medical procedures and improve lifestyle. However, in real life, physicians specialized in reproductive medicine often haven't sufficient available tools for offering personalized and adapted psychosocial care to patients involved in an ART process.

There are still gaps in the knowledge regarding individual perceptions and patient experiences of infertility and during the use of assisted reproduction technologies (ART). First, few studies have analyzed the entire impact of infertility and its management throughout the entire care pathway, namely, before, during and after ART procedures. Additionally, studies often focus on a specific aspect (i.e., sexual dysfunction [14]) and/or a time period in the reproductive life of couples: at the diagnosis of infertility, during the ART procedure [15], after ART drop out [16] or a long time after ART to study long-term outcome [17].Second, most studies focused on women, whereas there is little evidence regarding men's reported problems within the course of the infertility journey [18]. Third, most published work uses either standardized general or disease-specific tools, yet those instruments may fail to capture some dimensions of mental health, and they may also limit the spontaneous expression of affected people. Last, we lack real-life data regarding actual management and patient experience.

In France, in public infertility treatment Units, fertility treatments and ART are free of charge for women until 43 years. The government health insurance covers 6 intrauterine

inseminations and 4 IVF cycles per live birth, as well as the cost of absences from work for infertility treatment. For other health care costs (ex: psychologist, alternative and complementary medicine), patients can be refunded partially or totally by French government and private mutual insurers. In this context, the objective of our study was to collect perceptions of and real-life experiences of people treated for infertility with MAR treatments in a large sample of infertile French women and men throughout the entire process of their MAR program.

## Materials and methods

A prospective cross-sectional web-based survey was conducted by Ipsos, the largest French company in market and public opinion research, from October 7th, 2018, to October 28th, 2018, targeting French patients with a prior or current history of infertility in any specialized healthcare facility. The overall study sample has been previously targeted and investigated throughout the Ipsos Access panel; the study sample is composed of 314 077 people in France who are representative of the French population. Females and males in the targeted sample pool who were aged over 18 and under 50 years received an email invitation describing the study and directing the recipients to the secure anonymous survey website. Before taking part in the survey, participants were given a detailed description of the study and asked to provide consent to participate. Regarding the recruitment and targeting of the people participating in this survey, extensive quality procedures were in place to ensure that the survey inputs allowed for high quality survey outputs. Panel respondents were required to validate their registration via a security code to prevent automatic registration, and they double-opted in via email confirmation to ensure validity of the email address provided. All email extensions of clients, competitors, and Ipsos employees were removed, mismatched device settings and Geo-IP locations were also removed. Patterns in names, emails, and IP addresses collected at registration, and accounts that had multiple elements in common were removed.

Eligible respondents also had not participated recently in similar surveys. Strict panel usage rules were established to avoid interviewing the same people too often and to prevent them from being used too often for any individual type of survey. Duplicate device identification is also in place through digital fingerprinting (RelevantID©) and web/flash cookies. Respondents could only take the survey once, and this was assured by duplicate contact details identification. During the investigation, inattentive respondents were identified and removed. To identify someone who displayed inattentive survey-taking behavior through completing a survey too quickly, the time spent in the survey overall was measured, as well as the number of answers provided. This allowed us to calculate a completion speed, the number of answers provided per minute, for each respondent. To identify someone who displayed inattentive survey-taking behavior by providing identical answers across multiple questions within and across multiple grids, straight-lining response patterns were measured.

Inclusion criteria were established with the scientific committee of the study (BC, MG, AG, CS, EA). Medically Assisted Reproduction (MAR) is defined by the international Glossary on Infertility and Fertility care [1] and include "ovulation induction, ovarian stimulation, ovulation triggering, all ART procedures, uterine transplantation and intra-uterine, intracervical, and intravaginal insemination with semen of husband/partner or donor. However, we excluded from our survey women who underwent ovulation induction, because they couldn't report the burden of a laboratory intervention. People surveyed included males or females between 18 and 50 years old, currently undergoing a MAR procedure or having had an MAR procedure with or without a live birth (except those currently undergoing or having undergone MAR abroad).

The patient survey included a 56-point questionnaire developed for the study; the survey was completed and submitted online (See S1 and S2 Files: questionnaire available in French and in English). Because our objective was to provide an overview about the impact of infertility and MAR process throughout different aspects of the daily-life of patients, questions were elaborated by a scientific committee composed by physicians specialized in reproductive medicine, an Economic Doctor specialized in Human Resources and wellbeing at work (AL), and a representative (VR) from a National Infertility Association (bAMP) highly involved in France for supporting infertile couples. Before the survey, the questions were validated by infertile patients from bAMP Association. Questions were constructed both on existing literature (to validate data already described, as psychosocial impact) and on clinical experience reported both by physician and patients (as the professional impact, often described by patients).

The first part of the survey collected demographic and general information (age, sex, level of well-being, family status, history of MAR procedure or ongoing MAR procedure, type of MAR, number of attempts, etc.). The second part explored the respondents' personal history before resorting to MAR (delay before consultation, feeling when difficulties to conceive were first discovered, health professionals who made the diagnosis, etc.). The third and fourth parts of the survey included questions regarding experience with and perception of MAR (global feelings, psychological and physical impacts, impact on affective and sexual life, impact on relationships with others, impact on professional life, treatment burden, rapport with medical staff, expectations, etc.). The overall well-being score was self-reported by a numeric scale with rates ranging from 0 to 10 (0 for very low well-being to 10 for very high well-being); the impact of MAR on different domains of life was evaluated from 0 to 10 (0 for very low impact to 10 for very high impact). Most questions had multiple answers, and the results were expressed as a percentage of each answer.

## Data management and analyses

Statistical analyses and tests were performed using COSI software (M.L.I., 1994, France). Descriptive statistics include frequency tables, mean, standard deviations and 95% confidence interval (95% CI). A p value$<$0.05 was considered as significant for this statistical analysis. The current article focuses on highlighting the statistical differences between specific groups of population: people still undergoing MAR, people for whom MAR led to a live birth, people who dropped out of MAR, men and women. A system of letters has been implemented to illustrate the statistical differences between these subgroups. Letter B stands for people still undergoing MAR, letter C stands for people for whom MAR led to a live birth, letter D stands for People who dropped out of MAR, letter E stands for men and letter F stands for women. "+" refers to a superior significant difference.

This survey used anonymized patient data and was exempt from approval by an ethics committee according to the French national ethics law. Digital informed consent was obtained from all patients. A request for use of the database for research purposes was submitted to the French National Commission for Data Protection (*Commission Nationale de l'Informatique et des Libertés*).

## Results

### Demographic and general information

Among the 102,138 women and men of the targeted study sample by IPSOS, a total of 1,131 patients were recruited for the survey. Among them, 86 patients didn't answer questions and finally 1,045 patients (355 men; 690 women) were included. The characteristics of the studied population and of the three main subgroups are described in Table 1. Among all respondents,

**Table 1. MAR burden: A French national survey.** Characteristics of the whole population of respondents (n = 1045) and of the three main subgroups of the survey population (n = 942).

| | All n = 1045 (100%)[a] | People still undergoing MAR n = 199 (19%) (B) | People for whom MAR led to a live birth n = 522 (50%) (C) | People who dropped out of MAR n = 221 (21%) (D) |
|---|---|---|---|---|
| **Age (mean, 95% CI)** | | | | |
| All | 38.11 (37.5–38.7) | 32.8 (31.6–34) | 39.8 (39.1–40.4) | 40.7 (39.5–41.9) |
| Female | 38.1 (n = 689) (37.4–38.8) | 32.8 (n = 112) (31.4–34.2) | 39.5 (n = 360) (38.7–40.3) | 40.8 (n = 151) (39.3–42.3) |
| Male | 38.2 (n = 356) (37.2–39.2) | 32.9 (n = 87) (30.9–34.9) | 40.4 (n = 162) (39.1–41.7) | 40.5 (n = 70) (38.5–42.5) |
| **Age groups (n, %)** | | | | |
| 18–24 | 42 (4.0%) | 20 (10.1%) (+CD) | 8 (1.5%) | 6 (2.7%) |
| 25–34 | 262 (25.1%) | 105 (52.8%) (+CD) | 87 (16.7%) | 30 (13.6%) |
| 35–44 | 509 (48.7%) | 63 (31.7%) | 301 (57.7%) (+BD) | 100 (45.2%) (+B) |
| 45–50 | 232 (22.2%) | 11 (5.5%) | 126 (24.1%) (+B) | 85 (38.5%) (+BC) |
| **Marital status (n, %)** | | | | |
| Single | 100 (9.6%) | 11 (5.5%) | 44 (8.4%) | 31 (14%) (+BC) |
| In a couple | 945 (90.4%) | 188 (94.5%) (+D) | 478 (91.6%) (+D) | 190 (86.0%) |
| **Number of children under 18 at home (n, %)** | | | | |
| None | 297 (28.4%) | 78 (39.2%) (+C) | 18 (3.4%) | 155 (70.1%) (+BC) |
| 1 | 330 (31.6%) | 85 (42.7%) (+CD) | 173 (33.1%) (+D) | 39 (17.6%) |
| 2 | 321 (30.7%) | 31 (15.6%) | 253 (48.5%) (+BD) | 20 (9.0%) |
| 3 or more | 97 (9.3%) | 5 (2.5%) | 78 (15.0%) (+BD) | 7 (3.2%) |
| **Net mensual household income after deduction of income taxes (n, %)** | | | | |
| 1.250 € or less | 75 (7.2%) | 28 (14.1%) (+C) | 22 (4.2%) | 18 (8.1%) (+C) |
| 1.251 to 2.000 € | 164 (15.7%) | 42 (21.1%) (+C) | 62 (11.9%) | 39 (17.6%) (+C) |
| 2.001 to 3.000 € | 265 (25.4%) | 36 (18.1%) | 148 (28.4%) (+B) | 52 (23.5%) |
| More than 3.000 € | 436 (41.7%) | 76 (38.2%) | 237 (45.4%) | 84 (38%) |
| Refusal to answer | 105 (10.0%) | 17 (8.5%) | 53 (10.2%) | 28 (12.7%) |

(+B, +C, +D) p <0.05 Significant statistical superior differences between People still undergoing MAR, People for whom MAR led to a live birth and People who dropped out of MAR

[a] Among all respondents who weren't included in subgroups, 56 were pregnant, and 47 had not yet started MAR.

56 were pregnant, and 47 had not yet started ART. These 103 patients were not included in subgroups, because these subgroups were too small for being statistically analyzed. The remaining 943 patients were divided into three subgroups: people who succeeded in having a live birth following MAR (n = 522; 50%), those currently undergoing MAR treatment (n = 199; 19%), and those who dropped out of the MAR process without a live birth (n = 221; 21%). The mean time since drop out was 8.8 ± 4.9 years (95% CI: 7.7–9.9). Among this last subgroup, reasons for drop-out were reported as follows: personal decision (n = 82; 37%), discouragement by a too many failures (n = 66; 30%), taking a temporary MAR procedure break before another attempt (n = 39; 18%), financial reasons (n = 23; 10%), and medical reasons (n = 11; 5%).

Among patients who were currently undergoing MAR treatment (n = 199), the procedures were distributed as follows: intrauterine insemination (IUI) with sperm from the partner (n = 99; 50%), IUI with sperm from a donor (n = 21; 10%), in vitro fertilization (n = 64; 32%), and others (n = 15; 8%). The respondents had already experienced an average of 3.6 (95% CI:

3.3–3.9) MAR attempts among the whole study sample (n = 1045) and 3.9 (95% CI: 3.4–4.4) attempts among the patients who dropped out of the MAR process (n = 221). The mean general level of self-reported well-being was 6.7 (95% CI: 6.6–6.9) out of 10 and was significantly lower in people who dropped out of MAR (6.5, 95% CI: 6.1–6.8) vs the 2 other subgroups (p< 0.05).

## Personal history before resorting to a MAR procedure

The two main concerns prior to starting a MAR procedure were the risk of never becoming a parent (n = 593, 57%) and self-doubt about their own accountability in prior pregnancy failures (n = 485, 46%). Other findings are presented in Table 2. For 45% of respondents (n = 470), the first specialized consultation occurred less than 12 months after attempting natural conception. Among the population currently undergoing MAR, the patients had started their reproductive project 3.8 years (95% CI: 3.6–4.0) before participating in the survey.

## Perceptions of and relationship with the infertility healthcare system

Overall, the level of satisfaction with the infertility healthcare system was deemed high, with an average rating of 6.9 (95% CI: 6.7–7.0) out of 10, with 20% of respondents reporting a very high level of satisfaction (≥9 out of 10). However, satisfaction with MAR depended on the outcome; the people for whom MAR succeeded (n = 522) reported an average MAR satisfaction rating of 7.8 (95% CI: 7.6–7.9), while the average satisfaction rating was 5.3 (95% CI: 4.8–5.7) for people who dropped out of MAR (n = 221) (p< 0.05). Furthermore, the vast majority of patients expressed very high satisfaction regarding medical care received (Fig 1). The only

**Table 2. A French national survey.** MAR burden in the whole studied population (n = 1045) and in the three main subgroups of the respondents (n = 942). Responses to the question: "When you encountered first difficulties in having a child, what were all the questions you asked yourself at that time?".

| | All (%) | People still undergoing MAR n = 199 (19%) | People for whom MAR led to a live birth n = 522 (50%) | People who dropped out of ART n = 221 (21%) |
|---|---|---|---|---|
| | n = 1045 (100%) | (B) | (C) | (D) |
| Will **I ever have a child?** | n = 593 (57%) | n = 90 (45%) | n = 337 (65%) (+BD) | n = 119 (54%) |
| Is it my **fault?** | n = 485 (46%) | n = 85 (43%) | n = 259 (50%) | n = 104 (47%) |
| Will I have to **adopt?** | n = 371 (36%) | n = 62 (31%) | n = 201 (38%) | n = 86 (39%) |
| Is it my **partner's fault?** | n = 365 (35%) | n = 59 (30%) | n = 207 (40%) (+B) | n = 75 (34%) |
| Will I be eligible for **ART?** | n = 228 (22%) | n = 47 (24%) | n = 107 (20%) | n = 50 (23%) |
| Is it related to my **lifestyle?** | n = 197 (19%) | n = 39 (20%) | n = 99 (19%) | n = 40 (18%) |
| Is there a **l history** in my family or that of my partner's? | n = 203 (19%) | n = 38 (19%) | n = 106 (20%) | n = 41 (19%) |
| Is it related **to my weight?** | n = 176 (17%) | n = 44 (22%) (+D) | n = 87 (17%) | n = 28 (13%) |
| Is it **hereditary?** | n = 168 (16%) | n = 33 (17%) | n = 83 (16%) | n = 40 (18%) |
| Is it related **to my diet?** | n = 150 (14%) | n = 41 (21%) (+C) | n = 58 (11%) | n = 35 (16%) |
| Is it related to my **professional environment?** | n = 122 (12%) | n = 26 (13%) | n = 57 (11%) | n = 19 (10%) |

(+B, +C, +D) p <0.05 Significant statistical superior differences between People still undergoing MAR, People for whom MAR led to a live birth and People who dropped out of MAR

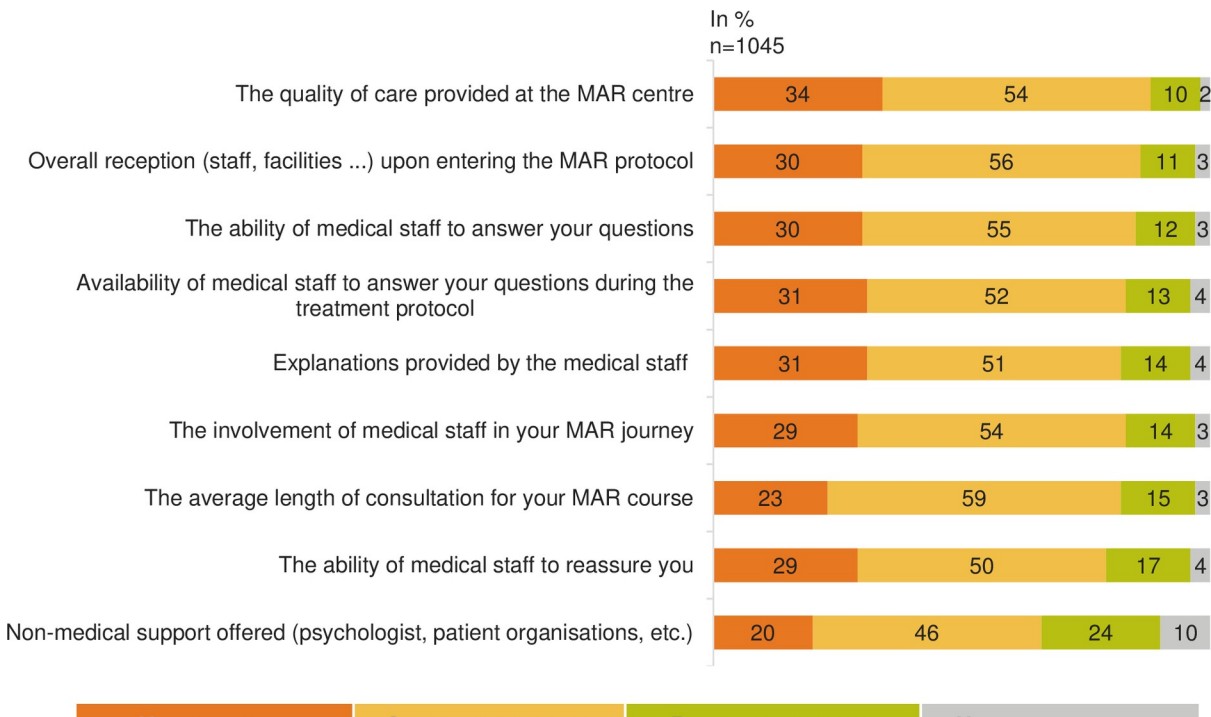

**Fig 1. Satisfaction concerning health care during the MAR program: A French national survey.** Percentage of answers to the question "Would you say that you have been very, somewhat, rather not or not at all satisfied with the following as part of your MAR program?" (n = 1 045).

rating that was substantially below the others was about non-medical support that was proposed (such as psychologists or patient support groups). The women and men surveyed did not show any significant differences in their feelings about and experiences with MAR.

Among people who had experienced a MAR procedure, 63% (n = 655) were offered an investigation of infertility as soon as they consulted for difficulties in conceiving a child. For 13% of them (n = 132), the gynecologist advised them to keep trying and come back later.

The paraclinical tests before starting MAR were deemed necessary (n = 972, 93%) and useful (n = 955, 91%) by the majority of surveyed people, but 75% also considered them stressful (n = 782) and long (n = 789). People currently on the MAR process (n = 199) were also likely to find these tests confusing (n = 126, 63%) and incomprehensible (n = 94, 47%). Otherwise, a majority of respondents (n = 187, 85%) among the subgroup of patients having dropped out of MAR management (n = 221) reported that the examinations needed for controlled ovarian stimulation (COS) monitoring were burdensome or very burdensome. One-third of the respondents (n = 63, 32%) currently on MAR process (n = 199) considered them very burdensome. Concerning the treatments for COS, the majority (75%, n = 149) thought that their current treatment of ovarian stimulation by gonadotropin injection had an impact on their daily life, 32% (n = 64) felt uncomfortable with self-injections and 31% (n = 62) thought that the treatment was hard to follow.

## Psychological, physical and social impact (Figs 2–5)

When responding to the question "What score between 1 and 10 would you give today to assess the psychological impact of your MAR care? (1 = you do not feel any psychological consequences and 10 means that you consider yourself to be very psychologically impacted)", the

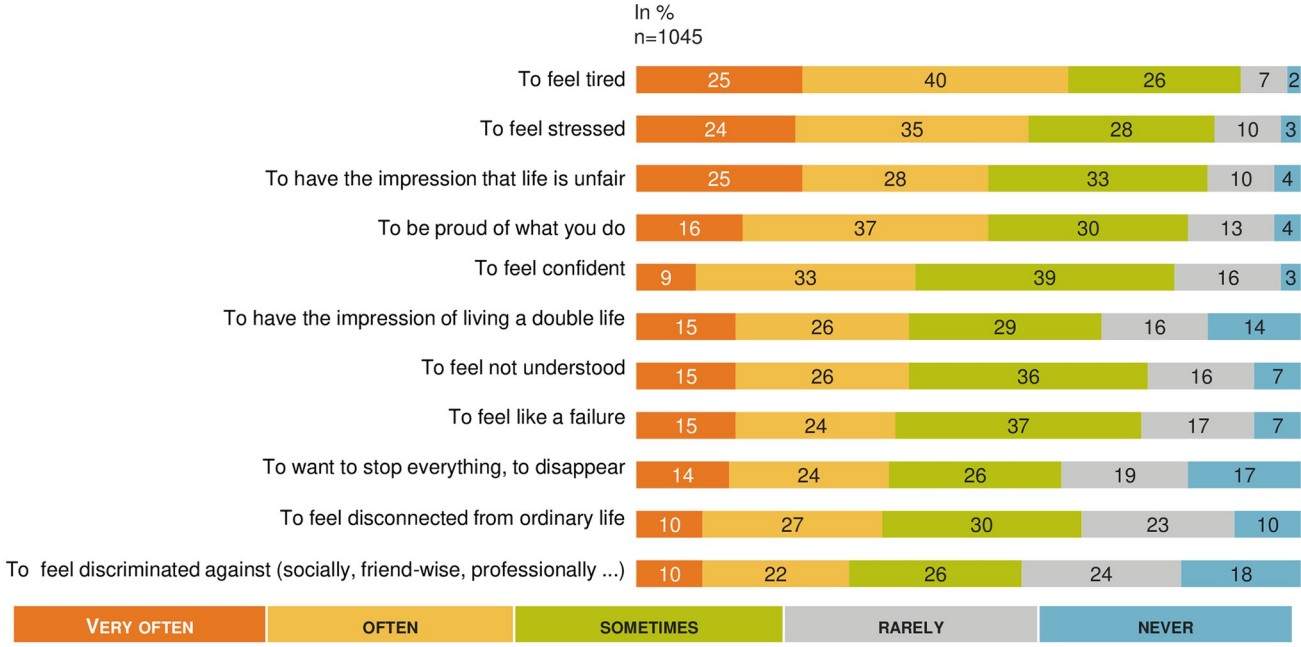

**Fig 2. MAR and psychological impact: A French national survey.** Percentage of answers to the question "Would you say each of the following happen to you very often, often, sometimes, rarely, or never?" (n = 1 045).

psychological impact was the most important impact of infertility and MAR process among the different areas explored (6.2, 95% CI: 6.0–6.4) with no significative difference between the 3 subgroups but a significant higher impact for women (6.4, 95% CI: 6.2–6.7) that for men (5.9, 95% CI: 5.5–6.2) ($p < 0.05$).

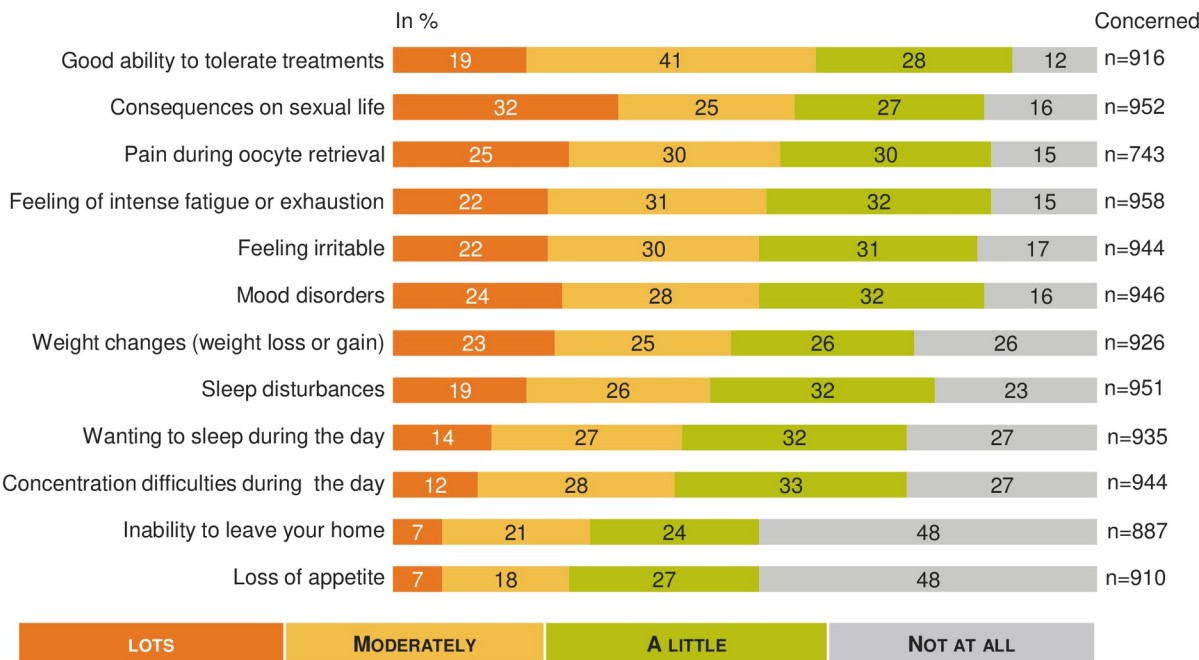

**Fig 3. MAR and physical impact: A French national survey.** Percentage of answers to the question "Regarding your physical condition in recent months, would you say that you felt each of the following a lot, moderately, a little, not at all?" (n = 916).

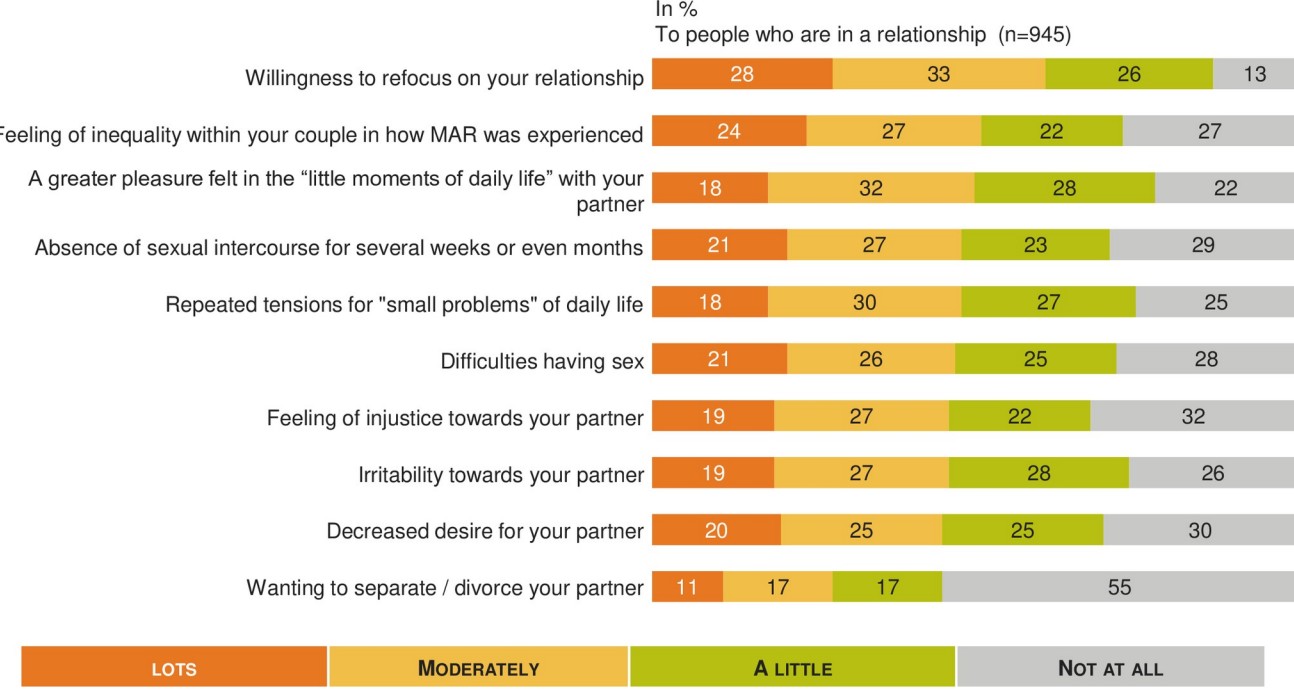

**Fig 4. MAR burden on affective life: A French national survey.** Percentage of answers to the question "Regarding your affective life would you say that you felt each of the following a lot, moderately, a little, not at all?" (n = 945).

The psychological impact was experienced strongly regardless of the situation in the MAR process (Fig 2). The majority (71%, n = 142) of the respondents currently on MAR process thought once a day or more often about their desire to have children. Among people who

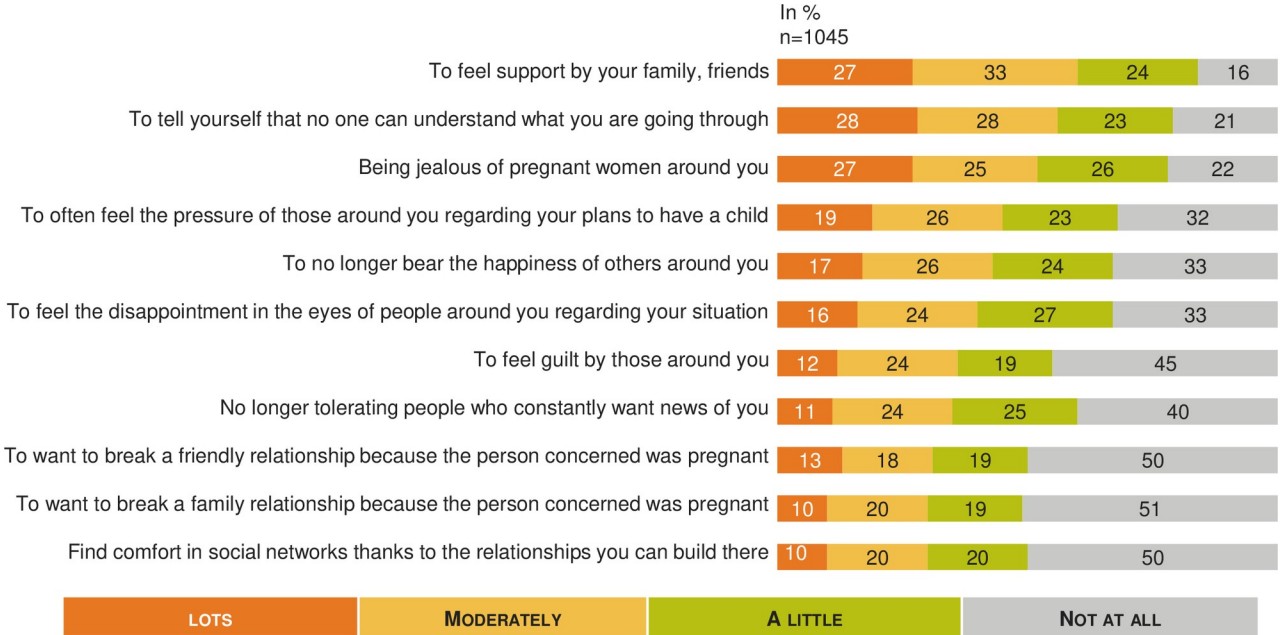

**Fig 5. MAR burden on social life: A French national survey.** Percentage of answers to the question "Regarding your relationship with those around you, would you say that you felt each of the following a lot, moderately, a little, not at all?" (n = 1 045).

dropped out of MAR without a baby, 30% (n = 66) thought about their desire to have children daily at the time of the survey. The score self-reported for evaluating the physical impact of infertility and MAR was rated 5.7 (95% CI: 5.4–5.9) out of 10, and this score wasn't significantly different between subgroups. However, physical impact was significantly increased for women: 5.8 (95% CI: 5.5–6.0) vs men 5.4 (95% CI: 5.0–5.7).

Fig 3 shows the percentage of response for each item and highlights the impact of MAR from moderate to a lot on sexual life (57%, n = 539; 93 not concerned), pain during transvaginal oocyte retrieval (55%, n = 410; 302 not concerned), intense fatigue or exhaustion (53%, n = 510; 87 not concerned), and mood disorders (52%, n = 491; 99 not concerned). Overall, 60% (n = 546; 129 not concerned) of respondents estimated they had a good ability to tolerate infertility treatments.

The self-reported impact on affective life was rated 5.7 (95% CI: 5.5–5.9) out of 10, with no significant difference between women and men ($p \geq 0.05$).

Different statements were tested regarding relationships within the couple (n = 945), and the results are summarized in Fig 4. Table 3 shows that impact on the couple's daily life was significantly increased in many areas for couples currently undergoing a MAR process

**Table 3. A French national survey.**

| | **All** | **People still undergoing MAR** | **People for whom MAR led to a live birth** | **People that drop out of MAR n = 190 (20%)** |
|---|---|---|---|---|
| | **n = 945 (100%)** | **n = 188 (20%)** | **n = 478 (51%)** | **(D)** |
| | | **(B)** | **(C)** | |
| **a lot + moderately[a] (n, %)** | | | | |
| **Willingness to refocus on your relationship** | n = 571 | n = 132 | n = 265 | n = 111 |
| | 60% | 70% (+CD) | 55% | 59% |
| **Feeling of inequality within your couple in how MAR was experienced** | n = 476 | n = 105 | n = 227 | n = 101 |
| | 50% | 56% (+C) | 48% | 53% |
| **A greater pleasure felt in the "little moments of daily life" with your partner** | n = 475 | n = 112 | n = 223 | n = 86 |
| | 50% | 60% (+CD) | 47% | 45% |
| **Absence of sexual intercourse for several weeks or even months** | n = 457 | n = 107 | n = 218 | n = 88 |
| | 48% | 57% (+CD) | 46% | 46% |
| **Repeated tensions for "small problems" of daily life** | n = 456 | n = 99 | n = 223 | n = 91 |
| | 48% | 53% | 47% | 48% |
| **Difficulties having sex** | n = 441 | n = 102 | n = 215 | n = 86 |
| | 47% | 54% (+C) | 45% | 45% |
| **Feeling of injustice towards your partner** | n = 431 | n = 109 | n = 188 | n = 91 |
| | 46% | 58% (+C) | 39% | 48% (+C) |
| **Irritability towards your partner** | n = 432 | n = 105 | n = 204 | n = 82 |
| | 46% | 56% (+CD) | 43% | 43% |
| **Decreased desire for your partner** | n = 419 | n = 92 | n = 199 | n = 86 |
| | 44% | 49% | 42% | 45% |
| **Wanting to separate / divorce your partner** | n = 271 | n = 75 | n = 95 | n = 68 |
| | 29% | 40% (+C) | 20% | 36% (+C) |

MAR burden on the affective couple's daily—live.(B,C,D) p <0.05: Significant statistical differences between people still undergoing MAR, people for whom MAR led to a live birth and people who dropped out of MAR.

(+B, +C, +D) p <0.05 Significant statistical superior differences between People still undergoing MAR, People for whom MAR led to a live birth and People who dropped out of MAR

[a]The % displays the subtotal of people mentioning "a lot" or "moderately" at each item. The statistical differences have been calculated on this subtotal.

(p < 0.05). Concerning the impact on sexual life, 21% (n = 202) of patients reported that infertility and MAR led to not having sexual intercourse for several weeks or even months. We didn't observe any difference between women and men, with 21.1% of women and 21.8% of men who reported having no sexual intercourse for several weeks (p≥ 0.05).

The average rating for the self-reported impact of MAR on relationships within social networks was 4.9 (95% CI: 4.6–5.1) out of 10 (n = 1045), with a significant higher impact (5.3, 95% CI: 4.8–5.8) or the subgroup of patients currently undergoing MAR (n = 199). When looking into the relationship with the social environment in more detail (Fig 5), the majority felt supported from moderate to a lot by friends and family (60%, n = 629), but they also often felt, from moderate to a lot, that no one was able to understand them (57%, n = 593) and developed jealousy towards pregnant women around them (52%, n = 546).

## Burden of MAR on professional life (Fig 6)

To the question "What score between 1 and 10 would you give today to assess the impact of MAR on your professional life? 1 = MAR had no impact on your professional life and 10 = you feel that your professional life is very impacted", the average rating for the self-reported impact of MAR on professional life was 4.8 (95% CI: 4.6–5.0) out of 10 with a significant higher impact (5.3, 95% CI: 4.9–5.7) for the subgroup of active worker patients currently in a MAR process compared to people for whom MAR led to a live birth (4.7, 95% CI: 4.5–4.9) and people who dropped out of MAR(4.4, 95%: 4.0–4.8) (p< 0.05). Overall professional impact was significantly increased in women (4.9, 95% IC: 4.7–5.1) compared to men (4.6, 95%: 4.3–4.9), (p< 0.05).

Among patients currently undergoing MAR (n = 199), 93% (n = 185) were active workers. Among them, 63% (n = 116) considered MAR to have had an impact on the organization of

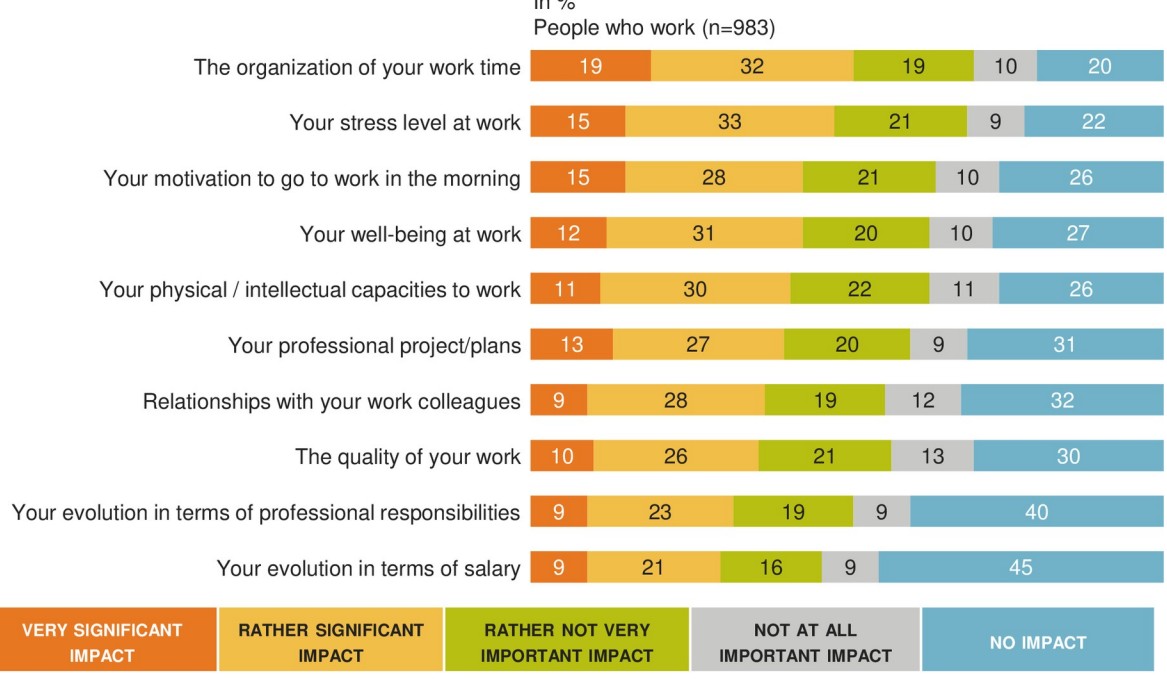

**Fig 6. MAR burden in professional life: A French national survey.** Percentage of answers to the question "Personally, do you feel that MAR has had a very significant, rather significant, rather not very important, not at all important impact or no impact on"" (n = 983).

**Table 4. MAR and professional burden for women and men who reported to have had a very significant and a rather significant impact on their professional life.**

| | All | Men | Women |
|---|---|---|---|
| | n = 984 | n = 347 | n = 637 |
| | (100%) | (35%) | (65%) |
| | | (E) | (F) |
| **Professional consequences** | | | |
| **n (%) Very+ rather important** | | | |
| The organization of your work time | n = 499 | n = 160 | n = 339 |
| | (51%) | (46%) | (53%) (+E) |
| Your stress level at work | n = 472 | n = 153 | n = 139 |
| | (48%) | (44%) | (50%) |
| Your motivation to go to work in the morning | n = 420 | n = 138 | n = 282 |
| | (43%) | (40%) | (44%) |
| Your well-being at work | n = 429 | n = 149 | n = 280 |
| | (44%) | (43%) | (44%) |
| Your physical / intellectual capacities to work | n = 402 | n = 138 | n = 264 |
| | (41%) | (40%) | (41%) |
| Your professional project/plans | n = 395 | n = 143 | n = 252 |
| | (40%) | (41%) | (40%) |
| Relationships with your work colleagues | n = 362 | n = 130 | n = 232 |
| | (37%) | (38%) | (36%) |
| The quality of your work | n = 351 | n = 135 | n = 216 |
| | (36%) | (39%) | (34%) |
| Your evolution in terms of professional responsibilities | n = 312 | n = 115 | n = 197 |
| | (32%) | (33%) | (31%) |
| Your evolution in terms of salary | n = 292 | n = 121 | n = 171 |
| | (30%) | (35%) (+F) | (27%) |

(+E, +F) p <0.05 Significant superior statistical differences between Men and Women

*The % displays the subtotal of people mentioning "very important" or "rather important" at each item. The statistical differences have been calculated on this subtotal.

their working time, and 51% (n = 95) reported that they were less motivated to go to work because of reduced well-being at work. Of the respondents, 55% (n = 101) reported a significant increase in their level of stress at work that was induced by infertility and MAR. Thus, 49% (n = 91) of patients felt that infertility and MAR had a significant impact on the quality of their work. On the other hand, 46% (n = 86) admitted that they had to lie to justify absenteeism necessary for the MAR procedure, and only 55% (n = 101) of them dared to use the absence authorizations provided by French law for people who underwent MAR. Finally, 35% of people (n = 65) stated that they preferred to resign as opposed to devote themselves fully to their career. Only 58% of respondents (n = 108) reported that they felt that their employer was understanding. Finally, 37% of them (n = 68) declared they had been pressured by management or colleagues during their career. This accumulation of negative reactions within the company led more than 35% of respondents (n = 64) to change employers. Table 4 reports the very important and rather important professional self-reported consequences of MAR. Professional impact was significantly more important (very important + rather important) for women concerning the work organization time management (p< 0.05).

## Discussion

Despite a universal health care for every infertile couple in France and despite a high level of satisfaction regarding French medical healthcare concerning MAR, our large national survey highlighted that both infertility and MAR treatments were associated with a major psychological burden with a negative impact throughout personal, social and professional life. Several perceived gaps were identified, thereby stressing areas for improvement.

Our findings confirmed prior research and identified precise issues deserving further investigations. The self-experience of infertility is often described by individuals and couples as a stressful condition and a heartbreaking situation, with anxiety and depressive symptoms with personal, partnership and social repercussions, that could decrease quality of life [19–24]. Infertility and ART also have marital consequences, inducing difficulties in partner communication [25] and sexual dysfunction [20,26]. In a meta-analysis, Mendoça et al. reported that lubrication, orgasm and satisfaction were mostly impaired in women [27]. In men, infertility has been reported to decrease self-esteem and sexual performance, with hypoactive sexual desire, erectile dysfunction and lack of sexual satisfaction [28,29]. In our study, 21.1% of women and 21.8% of men reported having no sexual intercourse for several weeks. Unfortunately, we haven't enough data to differentiate dysfunction and lack of sexual satisfaction. This result deserves a further specific study focusing on sexual dysfunction. Because ART permits to obtain a baby without sexuality, medical staff should be aware of the sexual quality of life in couples during the medical process and counsel them to maintain a satisfying relationship quality. In a meta-analysis, Frederiksen et al. concluded that psychological support could reduce psychological distress and improve pregnancy rates [30]. For Hämmerli et al., psychological interventions significantly increased pregnancy rates only in couples who were not receiving medical treatment, with an RR = 1.42, 99% CI: 1.02–1.99 [31]. We hypothesize that a better relationship quality could have a direct positive effect on sexuality and on spontaneous pregnancy rates. In our survey, 34% of responders pointed out the lack of nonmedical support during their ART program.

Our work raises several issues worth considering regarding infertility and its medical management. Some studies have explored quality of life (QoL) as a comprehensive indicator for the assessment of the psychological impact of complex clinical conditions, such as infertility. Infertile women have a worse QoL than both infertile men and fertile controls [21,32]. These findings also suggest that the period preceding the result of a treatment outcome can be considered a crucial moment for worsening psychological well-being relative to the beginning of ovarian stimulation and oocyte retrieval [32]. Furthermore, social, psychological and physical dimensions of QoL seemed to be more affected after ART failure [33].

We found similar results to those presented in the literature concerning the psychosocial impact of infertility and ART management [7,19,20,24,33]. Respondents appeared to be frequently impacted in many aspects of their life. Multiple concerns were reported, and numerous negative feelings and psychological impacts were raised. People were also somehow disrupted in their social interactions, and many respondents declared physical symptoms that were probably of multifactorial origin. Noticeably, most of those problems were reported regardless of the phase at which people were within the ART process, even for people who had dropped out of ART.

Women often complain of the burden of treatment, particularly in IVF. Physicians should adopt any available strategies to increase success of ART and reduce the risk of drop -out as well as the risk of complications, interruption of ovarian stimulation, and failure, such as the use of nomograms in the definition of gonadotropins doses for ovarian stimulation [34]. In this regard, the impact of procedure failure, particularly due to failed ovarian stimulation or

complication as Ovarian Hyperstimulation syndrome on the psychology and wellbeing of patients could be a further point of investigation.

While the consequences of infertility and ART on QoL, life satisfaction, marital life, sexuality and stress attitude have already been described, our study also pointed out the strong impact of ART on professional life, both on the individual working life of the respondents and on the enterprise environment. The work organization time management was a major concern, in particular for women, even in France, where the government health insurance covers the cost of absences from work for infertility treatment. To our knowledge, human resources teams have not yet embraced organizational support for decreasing the psychosocial burden of ART, which could induce depressive symptoms, anxiety, absenteeism, and job instability. It is possible that enterprise policies should be adjusted to help both women and men face ART logistics and help them combine professional and personal life during this difficult time. In France, enterprises probably underestimate that one in five people of reproductive age consult a fertility specialist at least one time in their life. Moreover, it has been reported that high-level occupational women often postpone childbearing until after age 35 and more often experience infertility related to age [35].

Our study has several strengths. First, it is a national survey including a very large number of women and men affected by infertility. Second, the fact that we included patients going through different stages of the ART process (successful ART, currently on ART and dropped-out of the ART process without a baby) allowed us to offer a global longitudinal view throughout the medical process. Third, the high number of questions, which were deeply anchored in day-to-day concerns, permitted us to retrieve concrete feedback that is thought to be directly actionable to implement countermeasures for quality of life and patient experience across the care cycle. Fourth. and in parallel with subjective insights, we collected real-life data regarding the actual management of patients at the national scale.

However, we recognize some limitations. The main limitation is that we have not used validated scales such as the FertiQol or Fertility Problem Inventory (FPI) for measuring specific patient outcomes [36]. However, validated scales are most useful for studying QoL changes in clinical interventional studies. In our study, we wanted to assess "real life" outcomes, with QoL outcomes determined from the real-life experience: patient outcomes and surveys have been determined by patients themselves (via a patient association, www.bamp.fr) and by experts following their professional experience. Our idea to build the survey was generated from the WHO definition of QoL. QoL is a wide concept for individuals, corresponding to "individuals' perception of their position in life in the context of the culture and value systems in which they live and in relation to their goals, expectations, standards and concerns" [37]. Therefore, the perspective of our work was to provide valuable information to physicians about the impact of infertility and ART on QoL and to help them to better support infertile couples in their everyday lives. One other limitation of our work is that we did not have medical data about the respondents, limiting the analysis of results. Indeed, many psychosocial factors could influence well-being and mental health during the ART process. For example, we do not know the etiology of the infertility of respondents. Massaroti et al. observed that women expressed more anxiety and general distress in cases of female infertility [38].

An online survey has some inherent biases. For instance, it limits study samples to people with internet access; these respondents are likely to be slightly different from a sociological standpoint than those without internet access. However, this means of surveying people, including patients, has dramatically grown over the last decade and is now considered valid if conducted according to predefined and established methods [39]. Second, the three subgroups of patients were quite unbalanced among the whole study sample; due to this and to the observational design of the study, no definitive interpretation can be drawn from the differences measured between subgroups for some items. In particular, a causal relationship cannot be

inferred with certainty between some baseline or care characteristics and subsequent outcomes. Third, the endpoints measurements were self-reported, increasing the risk of social desirability bias.

## Conclusions

Our national survey of a large sample of women and men showed a high overall level of satisfaction regarding the medical care received, even though some gaps could be found, particularly regarding a lack of nonmedical support. Nevertheless, our findings confirm that the burden of infertility and MAR treatments is not negligible, particularly in sexual, psychosocial and professional life. We suggest that the internal quality control program of each Reproductive Medicine center should include the monitoring of fertility-related quality of life throughout all the MAR procedures, as is done for the live-birth rate, multiple pregnancy rate, and ovarian hyperstimulation syndrome rate. Moreover, the QoL evaluations must consider both women and men. Reproductive Medicine centers should be encouraged to develop nonmedical support for all patients at any stage of infertility treatment. Enterprises should be warned of the professional impact of infertility to help employees reconcile personal and professional life.

## Supporting information

**S1 File.**
(DOCX)

**S2 File.**
(DOCX)

## Author Contributions

**Conceptualization:** Blandine Courbiere, Michael Grynberg, Anne Grelat, Virginie Rio, Elisangela Arbo, Céline Solignac.

**Data curation:** Céline Solignac.

**Formal analysis:** Blandine Courbiere, Arnaud Lacan, Michael Grynberg, Virginie Rio, Céline Solignac.

**Funding acquisition:** Elisangela Arbo, Céline Solignac.

**Methodology:** Céline Solignac.

**Project administration:** Elisangela Arbo.

**Resources:** Elisangela Arbo.

**Supervision:** Blandine Courbiere, Elisangela Arbo, Céline Solignac.

**Validation:** Blandine Courbiere, Arnaud Lacan, Michael Grynberg, Anne Grelat, Virginie Rio, Elisangela Arbo.

**Visualization:** Arnaud Lacan.

**Writing – original draft:** Blandine Courbiere, Céline Solignac.

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
