## [Decision Letter · Decision Letter 0]

8 Jun 2020

PONE-D-20-12026

The burden of assisted reproductive technologies (ART) on psychosocial and professional life: results from a French survey

PLOS ONE

Dear Dr. Courbiere,

Thank you for submitting your manuscript to PLOS ONE. After careful consideration, we feel that it has merit but does not fully meet PLOS ONE’s publication criteria as it currently stands. Therefore, we invite you to submit a revised version of the manuscript that addresses the points raised during the review process.

We look forward to receiving your revised manuscript.

Kind regards,

Antonio Simone Laganà, M.D., Ph.D.

Academic Editor

PLOS ONE

Journal Requirements:

4. Thank you for providing the following Funding Statement: 

YES,Financial support of the IPSOS survey was provided by Gedeon Richter France.

We note that one or more of the authors is affiliated with the funding organization, indicating the funder may have had some role in the design, data collection, analysis or preparation of your manuscript for publication; in other words, the funder played an indirect role through the participation of the co-authors.

If the funding organization did not play a role in the study design, data collection and analysis, decision to publish, or preparation of the manuscript and only provided financial support in the form of authors' salaries and/or research materials, please review your statements relating to the author contributions, and ensure you have specifically and accurately indicated the role(s) that these authors had in your study in the Author Contributions section of the online submission form. Please make any necessary amendments directly within this section of the online submission form.  Please also update your Funding Statement to include the following statement: “The funder provided support in the form of salaries for authors [insert relevant initials], but did not have any additional role in the study design, data collection and analysis, decision to publish, or preparation of the manuscript. The specific roles of these authors are articulated in the ‘author contributions’ section.”

If the funding organization did have an additional role, please state and explain that role within your Funding Statement.

Please also provide an updated Competing Interests Statement declaring this commercial affiliation along with any other relevant declarations relating to employment, consultancy, patents, products in development, or marketed products, etc.  

No authors have any competing interests that coud be perceived to bias this work.

We note that one or more of the authors are employed by a commercial company: Clinique Pasteur, Centre Mistral.

Please know it is PLOS ONE policy for corresponding authors to declare, on behalf of all authors, all potential competing interests for the purposes of transparency. PLOS defines a competing interest as anything that interferes with, or could reasonably be perceived as interfering with, the full and objective presentation, peer review, editorial decision-making, or publication of research or non-research articles submitted to one of the journals. Competing interests can be financial or non-financial, professional, or personal. Competing interests can arise in relationship to an organization or another person. Please follow this link to our website for more details on competing interests: http://journals.plos.org/plosone/s/competing-interests.

Additional Editor Comments (if provided):

The topic of the manuscript is interesting. Nevertheless, the reviewers raised several concerns: considering this point, I invite authors to perform the required major revisions.

Reviewers' comments:

Reviewer's Responses to Questions

**Comments to the Author**

1. Is the manuscript technically sound, and do the data support the conclusions?

Reviewer #1: Yes

Reviewer #2: Partly

Reviewer #3: Yes

Reviewer #4: Yes

2. Has the statistical analysis been performed appropriately and rigorously? 

Reviewer #1: N/A

Reviewer #2: No

Reviewer #3: Yes

Reviewer #4: N/A

3. Have the authors made all data underlying the findings in their manuscript fully available?

Reviewer #1: Yes

Reviewer #2: Yes

Reviewer #3: Yes

Reviewer #4: Yes

4. Is the manuscript presented in an intelligible fashion and written in standard English?

Reviewer #1: Yes

Reviewer #2: Yes

Reviewer #3: Yes

Reviewer #4: Yes

5. Review Comments to the Author

Reviewer #1: The manuscript “The burden of assisted reproductive technologies (ART) on psychosocial and professional life: results from a French survey” addressed impacts of infertility and assisted reproductive technologies (ART) throughout all aspects of life among infertile women and men, which is very interesting topic and rarely addressed before.

This study used a sample from an online survey which included 1,045 (355 men and 690 women) living or lived the experience of infertility and ART. This data provided an opportunity to evaluate the impact on quality of live for both men and women, which is initiative in ART associated researches.

The sampling method used by the authors is sound and represents the population, which is an information rich and has power for statistical test.

The authors provided detailed demographics of patients and patients’ responses for physical and mental problems asked in the questionnaire developed for the study in tables and figures, which helps readers to understand their results.

However, the authors didn’t provide statistical test results such that no solid evidences to support their discussion and conclusion. Following I would like to provide suggestions to improve this manuscript:

line 126; Descriptive statistics include frequency tables, means and standard deviations. I suggest use 95% Confidence interval to replace ± standard error.

line 137; Among all respondents, 56 were pregnant, and 47 had not yet started ART. These 103 patients not included in subgroups, which needs to indicate.

Line 147; intrauterine insemination (IUI) with sperm from the partner (n=99; 50%), IUI with sperm from a donor (n=21; 10%), those are fertility treatment but not ART since ART refers to IVF.

Line 148; an average 3.6±4.2, better to use mean and 95% confident intervals, noticed that 3.6 – 4.2 < 0, which makes no sense.

Line 150; The authors claim “The mean general level of self-reported well-being was 6.7±1.7 out of 10 and was not significantly different among the three subgroups.” I suggest providing P value to specify statistical significance.

Table 1; suggest use 95% confident intervals replace SD for continuous variable age, suggest use P values to specify statistical significance between sub-groups

Table 2; Suggest use P values to specify statistical significance between sub-groups

Line 171; “However, satisfaction with ART depended on the outcome of the ART; the people for whom ART succeeded (n= 522) reported an average ART satisfaction rating of 7.8±1.6, while the average satisfaction rating was 5.3±2.4 for people who dropped out of ART (n=221).” I suggest a statistical test should be performed to support this claim “satisfaction with ART depended on the outcome of the ART.” Also, not all patients had ART since some patients had fertility treatment such as IUI, which is not ART.

Line 174; Furthermore, the vast majority of patients expressed very high satisfaction regarding medical care received (figure 1). Not very clear here, again, missed statistical expressions to support this claim here. Figure 1 showed that very satisfy group less than 35% for all items. Most (>50%) are in the somewhat satisfied group for all items except the last, Aids/supports proposed to accompany you during your journey, which is 46%. It seems the authors grouped very satisfy patients and somewhat satisfied patients together for their claim here, which needs to indicate.

Line 177; Among people who had experienced ART, 63% (n= 655) were offered an investigation… to line 192, all listed % and numbers can’t be found anywhere else (tables or figures), which is confusion. Suggest make tables list those items to help readers understand

Line 209; Figure 3 shows the percentage of response for each item and highlights the impact of ART on sexual life (57%, n = 539; 93 not concerned). Suggest “… the impact of ART from moderate to a lot on sexual life…” to indicate this percent (57%) is a summation of both levels of impact (moderate and a lot). Same to the rest of description of figure 3 as well as figures 4, 5, and 6.

Line 270; In men, infertility has been reported to decrease self-esteem and sexual performance, with hypoactive sexual desire, erectile dysfunction and lack of sexual satisfaction [26,27]. In our study, 21% of patients reported having no sexual intercourse for several weeks. However, authors didn’t differentiate dysfunction and lack of sexual satisfaction between men and women though they did include both groups in their study.

Suggest performing statistic tests and provide P value for differences between subgroups of study population to support authors’ claim for their founding. P value should be provided in tables.

Suggest change the title as “The burden of fertility treatments and assisted reproductive technologies (ART) on psychosocial and professional life: results from a French survey” since this study includes patients who didn’t have ART.

Reviewer #2: Thank you for asking me to review this paper which reports findings from a study of the impact of infertility and assisted reproductive technology (ART) treatment on psychosocial wellbeing and the professional lives of women and men. While this is an important area of research, I have some reservations about this paper. I have the following comments and questions.

1) It is stated in the introduction that there is little evidence about the impact of infertility and assisted conception on psychosocial wellbeing. I disagree with this and refer to the comprehensive evidence presented in the ESHRE guidelines for psychosocial care in infertility and assisted conception:

Gameiro, S., Boivin, J., Dancet, E., de Klerk, C., Emery, M., Lewis-Jones, C., . . . Vermeulen, N. (2015). ESHRE guideline: routine psychosocial care in infertility and medically assisted reproduction - a guide for fertility staff. Human Reproduction, 30(11), pp. 2476-2485. doi:10.1093/humrep/dev177

2) In light of the evidence and the implications for practice presented in these guidelines I would argue that the current study only makes a very modest contribution to existing evidence.

3) We need some additional contextual information in the introduction to help readers. Who can have ART in France? What does it cost? Is it subsidised by government? Are there restrictions on how many cycles people can have?

4) It is stated that this was a 56 -item survey and the broad areas it covered are mentioned. I think much more information is needed about what the questions were based on (clinical experience? Existing literature?), if the questions had fixed choice response alternatives or if people gave their own responses. We also need an explanation for how satisfaction with the ‘ART health care system’ was measured. Was this one question or were respondents asked to state their satisfaction with different aspects of care? The methods section needs to detail what questions were asked and how responses were recorded.

5) The methods section should have a sub-section ‘Data management and analyses’ where the authors describe how data were grouped and analysed.

6) Were responses to questions (e.g. ‘When you first encountered difficulties in having a child, what were the questions that you asked yourself at that time?’) what the respondents answered or were they asked to choose from a list of response options? If they were fixed response options, how did the authors know that these were relevant? Were respondents given an ‘other’ option where they could say what questions they had asked themselves if they were not covered in the fixed response options?

7) The data are simply presented as frequency distributions. This makes the paper seem undigested. To understand what the data mean we at least need some statistical analyses to tell us if differences between groups are statistically significant and some univariate measures of association. Also, I think comparing women and men and reporting if they differ significantly in their responses would enhance the presentation of the data.

8) Relating to my previous point, for some of the data it is difficult to understand the rationale for presenting it by subgroup. What are readers supposed to make of the data in Table 2 for example? What does it mean if the proportions of people in the three subgroups differed in their choices of responses?

9) It is stated that the mean time since ‘drop-out’ was 8.5 years. What was the range? I presume this means that some had only recently ended treatment and others may have ended treatment more than a decade ago. It would be interesting to know if those who had ended treatment more recently differed in their responses from those who had moved on with their lives since ending treatment? After this the proportions stating various reasons for discontinuing treatment are reported. The percentages given look like proportions of the whole study sample? If so, this should be changed to proportions of those who had discontinued treatment without having had a baby.

Reviewer #3: The burden of ART is very well known since years.

However, in my opinion, studies analyzing the topic are always welcome to remember the clinical staff this important aspect of infertility.

The present study is very well done and simple to read. My decision is therefore to accept it despite the limitations underlined by the Authors.

Just few comments.

The impact of infertility on sexuality is described all the times and we know that it is present in the majority of couples. It is not difficult to understand why! My question is : there is any evidence that specific physiological approaches may avoid or solve this problem? Dealing with infertility couples since 35 years, my feeling is that it is very difficult to do it!

What it is instead terrible and requiring “ social” solutions is the impact on the professional organization. If I’m not wrong ,French is one of the few countries that formally declared infertility a social disease. Despite that, 35% of responders had to change employers. Can we image what happens in most of the other countries??

I believe that this part is the most interesting of the study and should be more analyzed in-depth because, otherwise from other psychological aspects, it is a negative impact that has to be solved by social interventions . And it is urgent to do it! It is very difficult to accept today that women, already facing all the phycological and physical impact related to infertility and ART, have to sacrifice the job and the career!

Reviewer #4: I was pleased to revise the manuscript entitled “The burden of assisted reproductive technologies (ART) on psychosocial and professional life: results from a French survey” (Manuscript Number: PONE-D-20-12026).

I was particularly pleased to review this paper. In my honest opinion, the topic is interesting enough to attract the readers’ attention. Methodology is accurate and conclusions are supported by the data analysis. Nevertheless, authors should clarify some point and improve the discussion citing relevant and novel key articles about the topic.

In general, the Manuscript may benefit from several minor revisions, as suggested below:

• All the text needs a minor language revision by a native English speaker person, in order to some typos, and grammatical errors.

• In the results it is unclear the response rate. I would suggest providing the actual number of patients who received the survey and patients who answered.

• I would suggest improving the discussion, at least briefly, highlighting the importance to adopt any available strategies to increase success of ART and reduce the risk of complications, interruption of ovarian stimulation, and failure, such as the use of nomograms in the definition of gonadotropins doses for ovarian stimulation (PMID: 30242498; PMID: 27835829) In this regard, the impact of procedure failure, particularly due to failed ovarian stimulation or complication, on the phycology of patient could be a further point of investigation.

• Regarding assisted reproductive techniques and psychological support, I would suggest discussing, at least briefly, to highlight its further importance when assisted reproductive techniques are adopt in the field of fertility preservation in oncological patients. (PMID: 32419847)

6. PLOS authors have the option to publish the peer review history of their article (what does this mean?). If published, this will include your full peer review and any attached files.

Reviewer #1: No

Reviewer #2: Yes: Karin Hammarberg

Reviewer #3: Yes: Anna Pia Ferraretti MD, PhD

Reviewer #4: No

---

## [Author Response · Author response to Decision Letter 0]

22 Jul 2020

Journal requirements

1.We attest that our manuscript meets PLOS ONE's style requirements. 

2. We included the questionnaire used in the study both in French and in English and we attest that it is not under a copyright (Please see the supplementary Supporting Information files)

3. Our data are available without ethical or legal restrictions, and as requested, we uploaded our anonymized data set on Mendeley data. The anonymized dataset has been registered under Courbiere Blandine (2020), “AMR dataset”, Mendeley data, V1, https://data.mendeley.com/datasets/6rmxbd56cf/1

4. Sorry for haven’t being accurate enough concerning the Funding Statement.

Financial support of the IPSOS Survey and the statistical analysis was provided by Gedeon Richter France. The funder provided support in the form of salaries for C. Solignac and E. Arbo but did not have any additional role in the study design, data collection and analysis, decision to publish, or preparation of the manuscript. The funder also assumed all the expenses for publication’s fees.The commercial affiliation with Gedeon Richter France doesn’t alter adherence of co-authors to PLOS ONE policies on sharing data and materials.

Specific roles of authors are better described in the ‘author contributions section:

- Pr B. Courbiere (BC) participated in the study design, questionnaire conception (56 items), data analysis, and manuscript preparation

- Dr A. Lacan (AL) participated in the data analysis and manuscript preparation

- Pr M. Grynberg (MG) participated in the study design, questionnaire conception and data analysis.

- Pr A. Grelat (AG) participated in the study design, questionnaire conception and data analysis.

- Dr C. Solignac (CS) participated in the study design, data analysis, decision to publish and preparation of the manuscript. 

- Dr E. Arbo (EA) participated in the study design, data analysis, decision to publish

BC and CS were leading for writing the manuscript and dealing with reviewer comments.

All authors belang to the scientific committee group and contributed equally to the survey, by drafting key questions of the questionnaire and synthesizing results.AL and VR report no conflict of interest. BC and MG reports consulting fees and speaker’s fees from Gedeon Richter France. AG reports consulting fees from Gedeon Richter France.

The commercial affiliation with Gedeon Richter France doesn’t alter adherence of co-authors to PLOS ONE policies on sharing data and materials.

5. Concerning Competing interests:

Dr Anne Grela MD, isn’t employed by Clinique Pasteur, Centre Mistral. “Clinique Mistral” is just the name of the ART Center where she practices as a private physician 

AL and VR report no conflict of interest. BC and MG reports consulting fees and speaker’s fees from Gedeon Richter France. AG reports consulting fees from Gedeon Richter France. Gedeon Richter provided salaries for C. Solignac and E. Arbo. The commercial affiliation with Gedeon Richter France doesn’t alter adherence of co-authors to PLOS ONE policies on sharing data and materials.

No authors have any competing interests that could be perceived to bias this work. 

Responses to Reviewers

 Reviewer #1: The manuscript “The burden of assisted reproductive technologies (ART) on psychosocial and professional life: results from a French survey” addressed impacts of infertility and assisted reproductive technologies (ART) throughout all aspects of life among infertile women and men, which is very interesting topic and rarely addressed before. This study used a sample from an online survey which included 1,045 (355 men and 690 women) living or lived the experience of infertility and ART. This data provided an opportunity to evaluate the impact on quality of live for both men and women, which is initiative in ART associated researches. The sampling method used by the authors is sound and represents the population, which is an information rich and has power for statistical test. The authors provided detailed demographics of patients and patients’ responses for physical and mental problems asked in the questionnaire developed for the study in tables and figures, which helps readers to understand their results.However, the authors didn’t provide statistical test results such that no solid evidences to support their discussion and conclusion. Following I would like to provide suggestions to improve this manuscript:

We are very grateful for the time you took to review our article and we thank you very much for your constructives commentaries that helped us to improve our manuscript.

All statistical analyses were performed with the COSI software for statistical data analysis. Now, descriptive statistical data include frequency tables, means and 95% Confidence Intervals.The description of data management and statistical analysis have been updated in the revised version of the manuscript.

> line 126; Descriptive statistics include frequency tables, means and standard deviations. I suggest use 95% Confidence interval to replace ± standard error.

Statistical analysis has been rigorously checked by a statistician, and as suggested, we updated the data in the text and in tables .

> line 137; Among all respondents, 56 were pregnant, and 47 had not yet started ART. These 103 patients not included in subgroups, which needs to indicate.

These two subgroups were too small for being analyzed as separate subgroups. This is indicated in legends of table and is now indicated in the manuscript.

> Line 147; intrauterine insemination (IUI) with sperm from the partner (n=99; 50%), IUI with sperm from a donor (n=21; 10%), those are fertility treatment but not ART since ART refers to IVF.

According to the French legislation, IUI is considered as an ART technique, as well as IVF and is submitted to the same (strict) reglementation. Only ovulation induction is considered as a “simple” fertility treatment. 

Medically Assisted Reproduction (MAR) is defined by The international Glossary on Infertility and Fertility care (Zegers-Hochschild et al., 2017) and include “ovulation induction, ovarian stimulation, ovulation triggering, all ART procedures, uterine transplantation and intra-uterine, intracervical, and intravaginal insemination with semen of husband/partner or donor. In consequence, following your commentary and because we hope that our article will be read by international readers, we modified, as suggested, the title of our manuscript: instead of “The burden of fertility treatments and assisted reproductive technologies (ART) on psychosocial and professional life: results from a French survey”, we propose “The burden of Medically Assisted Reproduction (MAR) on psychosocial and professional life: results from a French survey”.

The term ART has been withdrawn in the hole manuscript and has been replaced by MAR; 

> Line 148; an average 3.6±4.2, better to use mean and 95% confident intervals, noticed that 3.6 – 4.2 < 0, which makes no sense.

Standard error was replaced by 95% Confidence interval , as well as all the levels of satisfaction reported in the survey.For this result, we replaced 3.6 +/- 44.2 by 3.6 (95% CI: 3.3 to 3.9) 

> Line 150; The authors claim “The mean general level of self-reported well-being was 6.7±1.7 out of 10 and was not significantly different among the three subgroups.” I suggest providing P value to specify statistical significance.

We revised all the manuscripts and added P values when the result was significant.

> Table 1; suggest use 95% confident intervals replace SD for continuous variable age, suggest use P values to specify statistical significance between sub-groups

As suggested, we replaced SD by 95% confident intervals and added P values between sub-groups when the difference was significant.

> Table 2; Suggest use P values to specify statistical significance between sub-groups

A p value<0.05 was considered as significant. Adding P values between sub-groups was difficult for the readability of the tables. For a better readability, a system of letters has been implemented in tables to illustrate the statistical differences between the subgroups. Letter B stands for people still undergoing MAR, letter C stands for people for whom MAR led to a live birth, letter D stands for People who dropped out of MAR, letter E stands for men and letter F stands for women. “+” refers to a superior significant difference. We hope that we presented the results easier to understand.

> Line 171; “However, satisfaction with ART depended on the outcome of the ART; the people for whom ART succeeded (n= 522) reported an average ART satisfaction rating of 7.8±1.6, while the average satisfaction rating was 5.3±2.4 for people who dropped out of ART (n=221).” I suggest a statistical test should be performed to support this claim “satisfaction with ART depended on the outcome of the ART.” Also, not all patients had ART since some patients had fertility treatment such as IUI, which is not ART.

As suggested, we added P values between sub-groups when results were significant. The term ART has been withdrawn from all the manuscript and has been replaced by the term Medically Assisted Reproduction (MAR), that defines better the sample of responders of our survey.

> Line 174; Furthermore, the vast majority of patients expressed very high satisfaction regarding medical care received (figure 1). Not very clear here, again, missed statistical expressions to support this claim here. Figure 1 showed that very satisfy group less than 35% for all items. Most (>50%) are in the somewhat satisfied group for all items except the last, Aids/supports proposed to accompany you during your journey, which is 46%. It seems the authors grouped very satisfy patients and somewhat satisfied patients together for their claim here, which needs to indicate.

Sorry for haven’t being accurate enough. Indeed, for statistical analysis, we grouped very satisfied patients and somewhat satisfied patients together. It is now indicated in the text.

The description of data management and statistical analysis have been updated in the revised version of the manuscript.

> Line 177; Among people who had experienced ART, 63% (n= 655) were offered an investigation… to line 192, all listed % and numbers can’t be found anywhere else (tables or figures), which is confusion. Suggest make tables list those items to help readers understand

Because our survey scanned 56 aspects of a patient's daily lives, we chose to select the most relevant results to reflect the MAR burden on psychosocial and professional life. Some results that we evaluated as of greater interest are detailed in the tables or figures, others are just mentioned in the text. However, all the anonymized data are yet available in a supporting information file.

> Line 209; Figure 3 shows the percentage of response for each item and highlights the impact of ART on sexual life (57%, n = 539; 93 not concerned). Suggest “… the impact of ART from moderate to a lot on sexual life…” to indicate this percent (57%) is a summation of both levels of impact (moderate and a lot). Same to the rest of description of figure 3 as well as figures 4, 5, and 6.

Sorry for not being accurate enough. Indeed, for statistical analysis, we grouped “a lot and moderately “patients together. It is now indicated in the text for all the descriptions of figures.

> Line 270; In men, infertility has been reported to decrease self-esteem and sexual performance, with hypoactive sexual desire, erectile dysfunction and lack of sexual satisfaction [26,27]. In our study, 21% of patients reported having no sexual intercourse for several weeks. However, authors didn’t differentiate dysfunction and lack of sexual satisfaction between men and women though they did include both groups in their study.

We agree with you that it would have been interesting to assay the reasons why so many patients reported having no sexual intercourse for several weeks/months. We added your comment in the manuscript. We added also that we didn’t observe any difference between women and men concerning sexual burden. However, focusing on the sexual impact of MAR wasn’t the primary outcome of our survey.

Our study aimed to assess all the impacts of MAR on the personal, social and professional aspects of life. It was “a real-life study” with the objective to identify gaps for improving care of infertile patients. One of the frustrating limits of our study is to have observed some gaps, but we can’t explain them. All interesting data identified in this study must be studied in the future by specific studies focused on a specific area (sexual impact, professional impact..). 

> Suggest performing statistical tests and provide P value for differences between subgroups of study population to support authors’ claim for their founding. P value should be provided in tables.

We added the results of the statistical analysis in the tables. Statistical analysis has been entirely revised and methodology is better described in a specific subsection. 

> Suggest change the title as “The burden of fertility treatments and assisted reproductive technologies (ART) on psychosocial and professional life: results from a French survey” since this study includes patients who didn’t have ART.

We really apologize for the misemployment of the term ART. Instead, we should have employed the term Medically Assisted Reproduction (MAR), defined by The international Glossary on Infertility and Fertility care (2017) that include “ovulation induction, ovarian stimulation, ovulation triggering, all ART procedures, uterine transplantation and intra-uterine, intracervical, and intravaginal insemination with semen of husband/partner or donor. 

The term ART has been withdrawn from all the manuscript and has been replaced by MAR

Following your commentary, we changed the title of our article.

to: “The burden of Medically Assisted Reproduction (MAR) on psychosocial and professional life: results from a French survey”. We hope that you will be in agreement with this new title.

Reviewer #2: Thank you for asking me to review this paper which reports findings from a study of the impact of infertility and assisted reproductive technology (ART) treatment on psychosocial wellbeing and the professional lives of women and men. While this is an important area of research, I have some reservations about this paper. I have the following comments and questions.

Thank you very much for your interest in our article and for the time you spent on doing interesting comments, that helped us improve our manuscript.

1) It is stated in the introduction that there is little evidence about the impact of infertility and assisted conception on psychosocial wellbeing. I disagree with this and refer to the comprehensive evidence presented in the ESHRE guidelines for psychosocial care in infertility and assisted conception:

Gameiro, S., Boivin, J., Dancet, E., de Klerk, C., Emery, M., Lewis-Jones, C., . . . Vermeulen, N. (2015). ESHRE guideline: routine psychosocial care in infertility and medically assisted reproduction - a guide for fertility staff. Human Reproduction, 30(11), pp. 2476-2485. doi:10.1093/humrep/dev177

We added this major reference in the introduction. However, despite ESHRE guidelines concerning psychosocial care in infertility and medically assisted reproduction, our survey showed that psychosocial care is still insufficient or not adapted for a large part of infertile people. We agree with you that psychosocial care can reduce stress about medical procedures and improve lifestyle. However, in real life, physicians specialized in reproductive medicine often haven’t sufficient available tools for offering personalized and adapted psychosocial care to patients involved in an ART process.

2) In light of the evidence and the implications for practice presented in these guidelines I would argue that the current study only makes a very modest contribution to existing evidence.

With this survey, we chose to explore all the impacts of MAR on personal, social and professional life. We agree that some of our results are already known, as the impact on sexual life and psychosocial life. However, we identified a strong impact on the professional life that represents a gap still not known by employers and companies. As suggested by your colleague (Reviewer 3), we added some data in our results showing differences between women and men that need to be further studied.

3) We need some additional contextual information in the introduction to help readers. Who can have ART in France? What does it cost? Is it subsidised by the government? Are there restrictions on how many cycles people can have?

In our introduction, we added contextual information about the conditions of reimbursement of infertility treatments by the French Health System: 

In France, in public infertility treatment Units, fertility treatments and ART are free of charge for women until 43 years. The government health insurance covers 6 intrauterine inseminations and 4 IVF cycles per live-birth, as well as the cost of absences from work for infertility treatment. For other health care costs (ex: psychologist, alternative and complementary medicine), patients can be refunded partially or totally by French government and private mutual insurers. 

Despite this universal health care for every infertile French couple, the burden of medical procedures are high. 

4) It is stated that this was a 56 -item survey and the broad areas it covered are mentioned. I think much more information is needed about what the questions were based on (clinical experience? Existing literature?), if the questions had fixed choice response alternatives or if people gave their own responses. We also need an explanation for how satisfaction with the ‘ART health care system’ was measured. Was this one question or were respondents asked to state their satisfaction with different aspects of care? The methods section needs to detail what questions were asked and how responses were recorded.

Our objective was to provide an overview about the impact of infertility and MAR process throughout the daily-life of patients. Questions were elaborated by a scientific committee composed by physicians specialized in reproductive medicine, an Economic Doctor specialized in Human Resources and wellbeing at work, and representatives from a National Infertility Association (bAMP) highly involved in France for supporting infertile couples. Before the survey, the questions were validated by infertile patients themself.

Questions were constructed both on existing literature (to validate data already described, as psychosocial impact) and on clinical experience reported both by physician and patients (as the professional impact, often described by patients).

The questions had fixed choice response alternatives. The survey was completed and submitted online. The overall well-being score was self-reported by a numeric scale with rates ranging from 0 to 10 (0 for very low well-being to 10 for very high well-being); the impact of MAR on different domains of life was evaluated from 0 to 10 (0 for very low impact to 10 for very high impact).

We included the questionnaire used in the study both in French and in English (Please see the supplementary Supporting Information file)

> 5) The methods section should have a sub-section ‘Data management and analyses’ where the authors describe how data were grouped and analysed.

Sorry for haven’t being accurate enough and a specific sub-section concerning Data management and analyses have been added. 

Statistical analyses and tests were performed using COSI software (M.L.I., 1994, France). Descriptive statistics include frequency tables, mean, standard deviations and 95% confidence interval (95% CI). A p value<0.05 was considered as significant for this statistical analysis. The current article focuses on highlighting the statistical differences between specific groups of population: people still undergoing MAR, people for whom MAR led to a live birth, people who dropped out of MAR, men and women. A system of letters has been implemented in Tables to illustrate the statistical differences between these subgroups. Letter B stands for people still undergoing MAR, letter C stands for people for whom MAR led to a live birth, letter D stands for People who dropped out of MAR, letter E stands for men and letter F stands for women. “+” refers to a superior significant difference.

> 6) Were responses to questions (e.g. ‘When you first encountered difficulties in having a child, what were the questions that you asked yourself at that time?’) what the respondents answered or were they asked to choose from a list of response options? If they were fixed response options, how did the authors know that these were relevant? Were respondents given an ‘other’ option where they could say what questions they had asked themselves if they were not covered in the fixed response options?

Following your commentaries, the questions are available. We included the questionnaire used in the study both in French and in English (Please see the supplementary Supporting Information file)

> 7) The data are simply presented as frequency distributions. This makes the paper seem undigested. To understand what the data mean we at least need some statistical analyses to tell us if differences between groups are statistically significant and some univariate measures of association. Also, I think comparing women and men and reporting if they differ significantly in their responses would enhance the presentation of the data.

Statistical analysis has been rigorously checked by a statistician, and we updated the data in the text and in tables, as suggested.The description of data management and statistical analysis have been updated in the new version of the manuscript.

Moreover, we added comparisons between women and men when results were significantly different.

> 8) Relating to my previous point, for some of the data it is difficult to understand the rationale for presenting it by subgroup. What are readers supposed to make of the data in Table 2 for example? What does it mean if the proportions of people in the three subgroups differed in their choices of responses?

The current article focuses on highlighting the statistical differences between specific groups of population: people still undergoing MAR, people for whom MAR led to a live birth, people who dropped out of MAR, men and women. In table 2, a system of letters has been implemented to illustrate the statistical differences between these subgroups. Letter B stands for people still undergoing MAR, letter C stands for people for whom MAR led to a live birth, letter D stands for People who dropped out of MAR, letter E stands for men and letter F stands for women. “+” refers to a superior significant difference.A p value<0.05 was considered as significant for this statistical analysis.

> 9) It is stated that the mean time since ‘drop-out’ was 8.5 years. What was the range? I presume this means that some had only recently ended treatment and others may have ended treatment more than a decade ago. It would be interesting to know if those who had ended treatment more recently differed in their responses from those who had moved on with their lives since ending treatment? After this the proportions stating various reasons for discontinuing treatment are reported. The percentages given look like proportions of the whole study sample? If so, this should be changed to proportions of those who had discontinued treatment without having had a baby.

The mean time since drop out was 8.8 +/- 4.9 years.

We agree that it would be interesting to study if the time since ended treatment has an influence on the responses. However, analysing different subgroups in a sub-group of 221 patients would lack power for having significant results. For your information, please find below the details of these sub-groups. A specific study could further be of interest to assay the impact of infertility with the time in a population who dropped-out from medical procedures. 

For each subtitle, we describe if the percentages are given from people of the whole study sample or if they are given from patients issued from a specific subgroup. 

 Time from MAR drop-out All (%) n=221 (100%)

2 years or less : n= 33 (14.9%)

Between 3 and 7 yrs : n= 72 (32.6%)

Between 8 and 12 yrs: n= 58 (26.2%)

Between 13 and 18 yrs: n = 39 (17.6%)

More than 18 yrs n=19 (8.6%)

Reviewer #3: The burden of ART is very well known since years. However, in my opinion, studies analyzing the topic are always welcome to remember the clinical staff this important aspect of infertility. The present study is very well done and simple to read. My decision is therefore to accept it despite the limitations underlined by the Authors.

Thank you very much for the time you spent for reviewing our manuscript. We agree with you that it is important to remember the clinical staff the important burden of infertility and MAR. In 2015, in the ESHRE guideline, Gameiro et al. reported international recommendations for providing routine psychosocial care in infertility and medically assisted reproduction. However, in real life, physicians specialized in reproductive medicine often haven’t sufficient available tools for offering personalized and adapted psychosocial care to patients involved in an ART process. 

In France, despite an universal health care for every infertile couple, we observed that both infertility and MAR treatments were still associated with a major psychological burden with a negative impact throughout personal, social and professional life.

Just few comments.

> The impact of infertility on sexuality is described all the times and we know that it is present in the majority of couples. It is not difficult to understand why! My question is : there is any evidence that specific physiological approaches may avoid or solve this problem? Dealing with infertility couples since 35 years, my feeling is that it is very difficult to do it!

We agree with you that a lot of articles reported observations about the impact of infertility on sexuality. Despite guidelines for improving care in ART, we need interventional studies to study if some specific interventions could be useful for improving wellbeing and sexuality in couples during the stressful experience of infertility and MAR.In a meta-analysis, Frederiksen et al. concluded that psychological support could reduce psychological distress and improve pregnancy rates.

> What it is instead terrible and requiring “ social” solutions is the impact on the professional organization. If I’m not wrong ,French is one of the few countries that formally declared infertility a social disease. Despite that, 35% of responders had to change employers. Can we imagine what happens in most of the other countries??

In the introduction, we added contextual information about the conditions of reimbursement of infertility treatments by the French Health System. As you, we have been really surprised by these results. In France, the cost of absences from work for infertility treatment is covered by the government. Despite that, a lot of patients are shamed and don’t dare to tell it to their employers. To our knowledge, few is published about the professional impact of ART. Enterprises should be warned to help their employees reconcile personal and professional life.

> I believe that this part is the most interesting of the study and should be more analyzed in-depth because, otherwise from other psychological aspects, it is a negative impact that has to be solved by social interventions . And it is urgent to do it! It is very difficult to accept today that women, already facing all the phycological and physical impact related to infertility and ART, have to sacrifice the job and the career!

As suggested, we have developed the section about the professional impact of infertility treatments and added the table 4, comparing women and men.

Reviewer #4: I was pleased to revise the manuscript entitled “The burden of assisted reproductive technologies (ART) on psychosocial and professional life: results from a French survey” (Manuscript Number: PONE-D-20-12026).

I was particularly pleased to review this paper. In my honest opinion, the topic is interesting enough to attract the readers’ attention. Methodology is accurate and conclusions are supported by the data analysis. Nevertheless, authors should clarify some points and improve the discussion citing relevant and novel key articles about the topic.

Thank you very much for your interest for this article and for the time you spent reviewing it.

In general, the Manuscript may benefit from several minor revisions, as suggested below:

 • All the text needs a minor language revision by a native English speaker person, in order to some typos, and grammatical errors.

All the manuscript has been revised by an native English Speaker. Please find attached the American Journal Expert Certificate.

• In the results it is unclear the response rate. I would suggest providing the actual number of patients who received the survey and patients who answered.

We added these data in the Results section.

Among the 102 138 women and men between 18 and 50 years old of the targeted study sample by IPSOS (= the largest French company in market and public opinion research), a total of 1 131 patients were recruited for the survey. Among them, 86 patients didn’t answer questions and finally 1 045 patients (355 men; 690 women) were included. 

• I would suggest improving the discussion, at least briefly, highlighting the importance to adopt any available strategies to increase success of ART and reduce the risk of complications, interruption of ovarian stimulation, and failure, such as the use of nomograms in the definition of gonadotropins doses for ovarian stimulation (PMID: 30242498; PMID: 27835829) In this regard, the impact of procedure failure, particularly due to failed ovarian stimulation or complication, on the psychology of patient could be a further point of investigation.

As suggested, we added this point in the discussion.

Women often complain of the burden of treatment, particularly in IVF. Physicians should adopt any available strategies to increase success of ART and reduce the risk of drop -out as well as the risk of complications, interruption of ovarian stimulation, and failure, such as the use of nomograms in the definition of gonadotropins doses for ovarian stimulation (Di Paola et al., 2018) In this regard, the impact of procedure failure, particularly due to failed ovarian stimulation or complication as Ovarian Hyperstimulation syndrome on the psychology and wellbeing of patients could be a further point of investigation.

> • Regarding assisted reproductive techniques and psychological support, I would suggest discussing, at least briefly, to highlight its further importance when assisted reproductive techniques are adopt in the field of fertility preservation in oncological patients. (PMID: 32419847)

This article about the integrated gynaecological and psychological approach for the Fertility preservation in women affected by gynaecological cancer is very interesting. However, because we included in our survey only patients who underwent ART for infertility and not for fertility preservation, we are afraid to be out of scope in the discussion if we would add this reference. In case of fertility preservation for cancer, psychosocial impact is probably due more to the cancer itself than to the ovarian stimulation process, even if the FP procedure should be carefully performed to minimize burden in a high stressful context.

---

## [Decision Letter · Decision Letter 1]

27 Aug 2020

Psychosocial and professional burden of Medically Assisted Reproduction (MAR): results from a French survey

PONE-D-20-12026R1

Dear Dr. Courbiere,

We’re pleased to inform you that your manuscript has been judged scientifically suitable for publication and will be formally accepted for publication once it meets all outstanding technical requirements.

Kind regards,

Antonio Simone Laganà, M.D., Ph.D.

Academic Editor

PLOS ONE

Additional Editor Comments (optional):

I carefully evaluated the revised version of this manuscript.

Authors have performed the required changes, improving significantly the quality of the paper.

Reviewers' comments:

Reviewer's Responses to Questions

**Comments to the Author**

1. If the authors have adequately addressed your comments raised in a previous round of review and you feel that this manuscript is now acceptable for publication, you may indicate that here to bypass the “Comments to the Author” section, enter your conflict of interest statement in the “Confidential to Editor” section, and submit your "Accept" recommendation.

Reviewer #3: All comments have been addressed

Reviewer #4: All comments have been addressed

2. Is the manuscript technically sound, and do the data support the conclusions?

Reviewer #3: Yes

Reviewer #4: Yes

3. Has the statistical analysis been performed appropriately and rigorously? 

Reviewer #3: Yes

Reviewer #4: Yes

4. Have the authors made all data underlying the findings in their manuscript fully available?

Reviewer #3: Yes

Reviewer #4: Yes

5. Is the manuscript presented in an intelligible fashion and written in standard English?

Reviewer #3: Yes

Reviewer #4: Yes

6. Review Comments to the Author

Reviewer #3: I'm happy with the Authors responses to my comments and I have no additional comments. The paper is viable for publication

Reviewer #4: I was pleased to revise the manuscript entitled “Psychosocial and professional burden of Medically Assisted Reproduction (MAR): results from a French survey” (Manuscript Number: PONE-D-20-12026R1).

I was particularly pleased to review this paper. In my honest opinion, the topic is interesting enough to attract the readers’ attention. Methodology is accurate and conclusions are supported by the data analysis. Moreover, the authors addressed all the suggested revisions, and I appreciated the manuscript improvement.

7. PLOS authors have the option to publish the peer review history of their article (what does this mean?). If published, this will include your full peer review and any attached files.

Reviewer #3: **Yes: **Anna Pia Ferraretti

Reviewer #4: No

---

## [Editor Report · Acceptance letter]

14 Sep 2020

PONE-D-20-12026R1 

Psychosocial and professional burden of Medically Assisted Reproduction (MAR): results from a French survey 

Dear Dr. COURBIERE:

I'm pleased to inform you that your manuscript has been deemed suitable for publication in PLOS ONE. Congratulations! Your manuscript is now with our production department. 

Kind regards, 

on behalf of

Dr. Antonio Simone Laganà 

Academic Editor

PLOS ONE